# Country-Wide Ecological Health Assessment Methodology for Air Toxics: Bridging Gaps in Ecosystem Impact Understanding and Policy Foundations

**DOI:** 10.3390/toxics12010042

**Published:** 2024-01-05

**Authors:** Mohammad Munshed, Jesse Van Griensven Thé, Roydon Fraser, Bryan Matthews, Ali Elkamel

**Affiliations:** 1Department of Mechanical and Mechatronics Engineering, University of Waterloo, Waterloo, ON N2L 3G1, Canada; 2Lakes Environmental Software, Waterloo, ON N2L 3L3, Canada; 3Department of Chemical Engineering, Khalifa University, Abu Dhabi 127788, United Arab Emirates; 4Department of Chemical Engineering, University of Waterloo, Waterloo, ON N2L 3G1, Canada

**Keywords:** air toxics, ecological health assessment methodology, ecological screening quotient, food-chain multiplier

## Abstract

Amid the growing concerns about air toxics from pollution sources, much emphasis has been placed on their impacts on human health. However, there has been limited research conducted to assess the cumulative country-wide impact of air toxics on both terrestrial and aquatic ecosystems, as well as the complex interactions within food webs. Traditional approaches, including those of the United States Environmental Protection Agency (US EPA), lack versatility in addressing diverse emission sources and their distinct ecological repercussions. This study addresses these gaps by introducing the Ecological Health Assessment Methodology (EHAM), a novel approach that transcends traditional methods by enabling both comprehensive country-wide and detailed regional ecological risk assessments across terrestrial and aquatic ecosystems. EHAM also advances the field by developing new food-chain multipliers (magnification factors) for localized ecosystem food web models. Employing traditional ecological multimedia risk assessment of toxics’ fate and transport techniques as its foundation, this study extends US EPA methodologies to a broader range of emission sources. The quantification of risk estimation employs the quotient method, which yields an ecological screening quotient (ESQ). Utilizing Kuwait as a case study for the application of this methodology, this study’s findings for data from 2017 indicate a substantial ecological risk in Kuwait’s coastal zone, with cumulative ESQ values reaching as high as 3.12 × 10^3^ for carnivorous shorebirds, contrasted by negligible risks in the inland and production zones, where ESQ values for all groups are consistently below 1.0. By analyzing the toxicity reference value (TRV) against the expected daily exposure of receptors to air toxics, the proposed methodology provides valuable insights into the potential ecological risks and their subsequent impacts on ecological populations. The present contribution aims to deepen the understanding of the ecological health implications of air toxics and lay the foundation for informed, ecology-driven policymaking, underscoring the need for measures to mitigate these impacts.

## 1. Introduction

The authors’ previous research investigated the adverse effects of air toxics on human health [1]. During that research, it was identified that a noticeable gap exists in the broader literature concerning the ecological impact of these air toxics on terrestrial and aquatic ecosystems. Our previous study [1] recommended a country-wide ecological risk assessment to fill existing gaps in the research. As a consequence, this paper introduces the Ecological Health Assessment Methodology (EHAM). EHAM is based on the United States Environmental Protection Agency’s (US EPA’s) Screening Level Ecological Risk Assessment Protocol (SLERAP) for Hazardous Waste Combustion Facilities [2], expanding its application to a much wider range of emission sources. This paper focuses on estimating ecological risks only from industrial sources, and it should be noted that EHAM could include other significant emission sources, such as fugitive emissions or mobile sources if such data are available. The general procedure established by EHAM offers a flexible framework that can easily be adapted to integrate these additional sources, thereby broadening the scope of its applicability.

Given that the impact from air toxics emissions can be transported over vast areas, and given that food webs can extend over multiple ecological zones, improving the understanding of the sources of cumulative ecological (and health [3]) risk correspondingly requires that EHAM be applied over a vast area that captures the sources of interest. Furthermore, by applying EHAM over a vast area, more informed policy decisions can be made, such as determining where best to situate a new industrial emission source or air quality monitoring station. Ideally, the vast area would be worldwide, but this is impractical for computation, data availability, and political reasons, with data availability being the major technical barrier. In North America, even applying EHAM state- or province-wide remains difficult in many states and provinces due to a lack of emissions data. To emphasize the desirability of applying EHAM to a vast area, the authors sought to apply EHAM “country-wide”. Kuwait was selected, as it was the only country for which the authors had access to necessary, sufficiently large-scale (>10%), country-wide industrial air toxics emissions data. Do note that due to data confidentiality restrictions, specific information identifying the sources cannot be disclosed. This is not a problem for this paper, which focuses on presenting the method to implement EHAM; however, using EHAM for policy decisions may have to confront source confidentiality considerations. Kuwait has two major air toxics-vulnerable ecosystems: desert and coastal ecosystems.

In the present study, the authors adapted air modeling and deposition simulations, originally conducted for human health risk assessments, to estimate ecological risks associated with air toxics. While the case study concentrates on Kuwait’s unique desert and coastal ecosystems, the methodologies implemented and the insights gained hold broader implications given the global importance of diverse ecosystems under threat from air toxics.

The fate and transport of air toxics within ecological settings exhibit characteristics similar to those impacts observed in human health assessments, as extensively detailed in [4]. This similarity highlights the interconnectedness of human and ecological health, thereby broadening our comprehension of the extensive implications of air pollution.

Ecosystems, fundamental in maintaining global biodiversity and health, play distinct roles in our environment. Wetlands, often called the “kidneys of the Earth” [5], not only purify our water but, like other ecosystems, also suffer from air toxics infiltration. Similarly, grasslands and forests maintain soil fertility, with their root systems acting as defenders against erosion [6,7]. Although it is common to think of wetlands, grasslands, and forests when speaking of ecosystems, and to think of deserts as barren, desert ecosystems such as those that dominate Kuwait do play a crucial environmental role as carbon sequestration sinks due to their vast size [8]. Furthermore, Kuwait has coastal ecosystems, characterized by their unique interface between land and sea, that are vital for biodiversity conservation and that provide crucial services like storm protection and nursery grounds for many marine species [9]. Coastal ecosystems are particularly vulnerable to air toxics that can accumulate in the coastal food web, impacting both terrestrial and marine life. For example, airborne mercury, primarily from industrial sources, can be deposited in water bodies, converting to methylmercury in aquatic systems. This toxic form then bioaccumulates in fish, ultimately impacting birds and mammals that rely on these fish for food. Furthermore, the influence of air toxics on one ecosystem can have ripple effects, altering nutrient cycles and energy flow in adjacent ecosystems. This intricate web of interactions necessitates a comprehensive, multi-ecosystem-based approach to manage and mitigate the effects of air pollution.

A summary of the entry mechanisms and affected trophic levels of air toxics in both terrestrial and aquatic ecosystems is provided in Table 1.

To complement and clarify the data in Table 1, Figure 1 shows the multi-dimensional ecological ramifications of air toxics across both terrestrial and aquatic ecosystems. Figure 1 highlights the impacts of air toxics on diverse aquatic organisms, from benthic invertebrates and vegetation to higher-order predators like fish and amphibians. Similarly, in terrestrial ecosystems, the figure demonstrates how soil-deposited pollutants can have cascading effects, influencing a range of organisms from vegetation to herbivores like deer and rabbits, as well as predators such as owls and red foxes. Table 1 and Figure 1 provide an easily interpretable overview of the ecological risks associated with air toxics. These elements give readers both the data and the conceptual framework needed to understand the broader implications.

In summary, EHAM as described here may provide a structured and adaptable framework for assessing the ecological risks posed by air toxics at country-wide level. Although this study employs the methodology within the specific environmental context of Kuwait, the principles and approaches are adaptable to other ecosystems. This adaptability enhances the methodology’s generalizability, offering a valuable tool in the global endeavor to mitigate ecological harm from air toxics.

### Literature Review, Objectives, and Additional Contributions

This section provides a literature review focused on the effects of air toxics on ecological health and highlights that past ecological risk assessments have primarily concentrated on chemicals other than air toxics. Additionally, this section outlines the research objectives and the contributions of the current study.

Examples of significant impacts from air toxics on select organisms are given in Table 2. Note that these impacts in Table 2 are all determined by exposing an organism to varying concentrations of a toxic, and may be used to generate toxicity reference values (TRVs) used by EHAM, as discussed in Section 2.8. That is, these impacts are not ecological risk determinations, but rather organism risk determinations. The varied examples in Table 2 also show the diverse impacts that air toxics can have on ecological receptors and help to contextualize the broader ecological implications of air pollution.

In addition to the direct impacts mentioned earlier, bioaccumulation (gradual accumulation of air toxics within organisms over time) and biomagnification (increase in concentration of air toxics as they move through different trophic levels) contribute to the amplification of ecological impacts. Notably, phytoplankton and mesozooplankton, which form the base of the food chain, play a critical role in transferring the effects of air toxics upward within the food web. This cascading effect ultimately influences aquatic and terrestrial apex predators. The widespread consequences of air toxic contamination in complex ecosystems thus become evident.

While the primary focus of this research investigates the impacts of air toxics on ecosystems, it is also important to acknowledge that these effects can indirectly impact wildlife through the consumption of contaminated plants, fish, or meats. This pattern of contamination and exposure follows a trajectory similar to human exposure to air toxics’ fate and transport, as extensively detailed in the authors’ previous publication [55]. The intake of such contaminated food escalates the risks of exposure for animals, emphasizing the necessity of robust air quality standards in controlling toxic emissions and mitigating ecosystem contamination.

Table 3 reveals how the current work contributes to the literature on the effects of toxics on ecosystems, and although the list of references in Table 3 is not all-inclusive, it does reflect the scarcity of studies into the impact of air toxics on ecosystems. The distinguishing features used to characterize the literature presented in Table 3 follow the Source, Pathway, and Receiver model, which parallels the workflow of EHAM. The Source, Pathway, and Receiver model [56] is a framework used to understand and assess pollution’s impacts. It traces the journey of an air toxic from its origin (“Source”), through the routes it takes in the environment (“Pathway”), to the entities that may be affected or harmed (“Receiver”). This model offers a perspective on how air toxics interact with and impact ecosystems, providing insights for risk assessments.

Most prior ecological risk assessments have focused on chemicals other than air toxics, such as pesticides. Regulatory frameworks like the Federal Insecticide, Fungicide, and Rodenticide Act (FIFRA) [70] have propelled research chiefly toward pesticides. Similarly, federal laws such as the American Clean Water Act [71] and the Endangered Species Act [72] have mainly directed attention toward waterborne toxics. In Europe, the European Food Safety Authority (EFSA) has established risk assessment frameworks for human health, animal health, and ecological areas [73] that are similar to the EPA protocols. To date, the EFSA frameworks have primarily been applied to plant protection products, i.e., pesticides, insecticides, and fungicides [74]. These assessments, as substantiated by the literature reviewed in Table 3, are invaluable in their targeted scope but frequently limited in geographic extent and seldom include air toxics. This narrow approach calls for a more comprehensive evaluation that considers air toxics and encompasses both terrestrial and aquatic ecosystems on a country-wide scale. In addition to air toxics, persistent organic pollutants (POPs) are noteworthy due to their environmental persistence and potential for bioaccumulation [75]. While POPs are not the primary focus of this study, their prospective integration into the EHAM presents a significant research avenue. The primary objective of this work is twofold: firstly, to introduce the Ecological Health Assessment Methodology, a novel and comprehensive approach for ecological risk assessment of air toxics that transcends traditional methods; and secondly, while EHAM is adept for country-wide evaluations, its design also facilitates detailed ecological risk analyses across various regions within a country, allowing for comparative assessments at multiple geographical scales.

#### Additional Contributions

Developing unique food web models tailored to local ecosystems to achieve a country-wide ecological risk assessment. These models incorporate region-specific trophic interactions, allowing for a more accurate and localized risk assessment;Extending US EPA methodologies originally created for hazardous waste combustion facilities to include other emission sources like wastewater treatment plants and glycol dehydration units;Estimating the country-wide cumulative ecological risk from simultaneous exposure to numerous air toxics such as polycyclic aromatic hydrocarbons originating from multiple industrial sources.

## 2. Materials and Methods

In alignment with the stated objectives, this section outlines the methodology devised to fill the identified gaps in prior ecological risk assessments. The methodology was validated using accepted emissions estimation methods and algorithms developed by reputable organizations, such as the US EPA and the American Petroleum Institute (API) [1]. Furthermore, air dispersion modeling was performed using AERMOD, a steady-state Gaussian plume model, which is the most extensively validated model for the present purpose and is currently the US EPA’s preferred regulatory model [76]. The Ecological Health Assessment Methodology (EHAM) is outlined in the following sequential steps:(1)Develop a country-wide emissions inventory by identifying and quantifying the sources of air toxics;(2)Perform air dispersion and deposition modeling using current US regulatory models. This step aims to estimate the atmospheric pollutant concentrations and their annual deposition rates, focusing on habitat-specific scenario locations within the defined project area (e.g., country);(3)Characterize exposure settings for affected habitats within the project area;(4)Develop habitat-specific food webs based on exposure data and receptor interactions;(5)Select assessment endpoints within each trophic level of the habitat-specific food web;(6)Identify measurement receptors to assess air-toxics-related measures of effect, such as toxicity values or receptor-specific chronic no-observed-adverse-effect levels (NOAELs);(7)Assess exposure via direct uptake and ingestion, targeting both lower- and higher-trophic-level receptors based on their interaction with air toxics;(8)Evaluate the toxicity of air toxics by identifying toxicity reference values;(9)Estimate the country-wide cumulative ecological risk from simultaneous exposure to numerous air toxics from multiple industrial sources.

### 2.1. Emissions Inventory

In the first step of the methodology, a country-wide emissions inventory is developed to identify and quantify the sources of air toxics. For this study, Kuwait served as the project area and was divided into three distinct air quality zones. In areas with varying wind patterns and urban pollution sources, further considerations may be needed, such as subdividing the area based on similar air quality, meteorological conditions, and pollution source types.

It is important to acknowledge the uncertainties in emission estimates, which include variability in emission factors, inaccuracies in activity data, and challenges posed by temporal and spatial resolution. Additionally, differences in emissions during start-up and shut-down phases compared to steady-state conditions must be considered.

This step establishes the base data layer for ecological risk assessment. The completeness and availability of the emissions inventory are essential for accurately evaluating the ecological risks associated with exposure to air toxics. Though beyond the scope of this paper, the existing emissions inventory can be augmented to include on-road vehicle emissions—a detailed method for estimating these emissions is thoroughly explained in a separate research paper [55].

For additional details on the methods and algorithms employed in emissions estimation, readers are referred to “Section 2.1” in the authors’ work in [1].

### 2.2. Air Dispersion and Deposition Modeling

The air dispersion and deposition modeling for this study was performed using AERMOD, which is the current regulatory model preferred by the US EPA, employing its version 22112 [76]. Further details can be found in the authors’ previous work on human health, as detailed in [1], specifically in “Section 2.2”.

### 2.3. Exposure Setting Characterization

In this step, the focus is on assessing the potential cumulative ecological impact due to the release of air toxics from multiple facilities throughout the country. This involves two primary processes: the identification of ecological receptors and the selection of habitats.

For the selection of habitats, environments potentially affected by facility emissions across the diverse landscapes of the country are identified by correlating them with the air dispersion modeling results. These results consist of the unitized output of concentration and deposition rates. Habitats are characterized by their biotic and abiotic elements and typically fall into two main categories: terrestrial and aquatic. Tools like land use land cover (LULC) maps, topographic maps, and aerial photographs are used to delineate these habitats accurately.

Within the exposure setting characterization, terrestrial habitats across the country are chosen based on air parameter values from AERMOD. The process starts with defining the habitats to be evaluated and then identifying specific receptor grid nodes with the highest yearly average air concentration and deposition values. The proximity of several water bodies, which influence receptor intake values in exposure equations, is also considered.

For aquatic habitats, the distinction between a range of water bodies and non-submerged areas across the country is crucial. The selection process mirrors terrestrial methods but with an added layer of considering air toxics loading from watersheds. For example, an oil and gas facility located upstream might release air toxics such as PAHs, which are particle-bound, like benzo(a)pyrene. When deposited, these air toxics could travel through the watershed and accumulate in downstream aquatic habitats. This downstream location, even if distant from the emission source, might experience increased levels of these air toxics due to runoff and water flow, thereby emphasizing the need to evaluate not just direct air deposition, but also the indirect introduction of air toxics through the watershed.

The selection process involves defining the aquatic habitats and pinpointing receptor grid nodes, particularly those with the highest annual average concentrations. Special attention is given to large water bodies, focusing closely on the most impacted areas and employing additional or nearest receptor grid nodes when needed.

The process of identifying ecological receptors ensures a representation of both plant and animal communities within a habitat. Ecological receptors are selected based on their known or projected sensitivities to air toxics, their ecological importance as keystone or indicator species, and their direct relevance to human health (e.g., species consumed by local populations).

This identification underscores all potential exposure pathways, including the food chain. The food web specific to potentially impacted habitats is explicitly defined to ensure that complete exposure pathways for both plant and animal communities are ascertained.

This research employed distinct techniques depending on the habitat’s nature, whether terrestrial or aquatic. For instance, analysis of aquatic habitats might involve the use of hydrological models to estimate concentrations of air toxics in different water layers, while that of terrestrial habitats may utilize soil sampling and bioaccumulation models to gauge the levels of air toxics in the soil and its subsequent impact on plant and animal life. Analysis of aquatic habitats delves deeper into water quality metrics, whereas that of terrestrial habitats emphasizes soil integrity, vegetation cover, and air-toxic sensitivities. This methodology encompasses defining boundaries using the air dispersion model’s receptor grid nodes, evaluating the highest concentrations of air toxics, and understanding the impact on water bodies and watersheds across the country.

### 2.4. Habitat-Specific Food Web Development

Using data from the exposure setting characterization, habitat-specific food webs are constructed to represent guilds and communities of receptors exposed to air toxics emissions from various facilities throughout the country. In this context, a guild refers to a group of species that function at a specific trophic level and exploit a common resource base in a similar way [77]. To organize the structure of these food webs, trophic levels are employed. The first trophic level (TL1) includes primary producers like plants. The second trophic level (TL2) consists of primary consumers, such as herbivores and detritivores. Higher trophic levels include omnivores and carnivores, classified as the third trophic level (TL3) and fourth trophic level (TL4), respectively. TL3 and TL4 organisms typically consume a variety of food sources, including those from the lower trophic levels.

Exposure pathways in these habitat-specific food webs are categorized into two primary types: direct and indirect exposure pathways. In this study, a direct exposure pathway refers to the immediate transfer of air toxics to an ecological receptor through avenues such as respiration or direct contact with contaminated media (e.g., soil, water). Conversely, an indirect exposure pathway involves a more complex transfer mechanism, typically through the food web itself. For example, air toxics may be absorbed by primary producers (TL1) and subsequently bioaccumulate up the trophic levels, affecting primary consumers (TL2), omnivores, and carnivores (TL3 and TL4). The criteria for categorizing these pathways are based on the number of intermediary steps between the release of air toxics and their contact with the ecological receptor, as well as the mechanisms of transfer (e.g., ingestion, respiration, dermal absorption).

Food webs show how energy, originating from primary sources like plants, is transferred through different organisms. The effectiveness of these webs in revealing the exposure pathways hinges on factors such as the dietary habits of organisms, the specific entities within the chain, and aspects like the availability and breakdown of the assessed air toxics. These habitat-specific food webs are important in highlighting both direct and indirect exposure pathways. Furthermore, they facilitate the establishment of mathematical relationships between guilds (e.g., quantification of the daily dose of air toxics ingested by a measurement receptor) and provide a detailed exposure analysis for different ecological receptors. Utilizing the community approach, a prevalent framework in the field [2], the following steps are integral to food web development and embody the core elements of this approach:Within a habitat, potential receptors are identified and grouped based on their feeding habits or guilds;The food web structure is organized by trophic levels, such as plants and primary consumers (herbivores);An important component involves delineating the feeding relationships among distinct guilds and communities.

The completion of these steps, according to the community approach, results in a thorough food web for each assessed habitat, designed for utilization in subsequent risk characterization.

### 2.5. Habitat-Specific Trophic Level Assessment Endpoint Selection

An assessment endpoint serves as an explicit representation of an ecological attribute requiring protection [78], particularly in the context of food web structure and ecosystem function. These assessment endpoints are identified on a granular level, specific to each community and guild within individual trophic layers of habitat-specific food webs. Guild-specific examples include the roles of seed dispersers and decomposers, while community-specific examples focus on attributes like species richness and productivity. The careful selection of these endpoints aims to ensure the overall ecological integrity of both guilds and communities, thereby safeguarding the structure and function of the habitat-specific food webs they comprise.

### 2.6. Measurement Receptor Identification for Measures of Effect Assessment

Identifying measurement receptors is critical when evaluating the ecological risk associated with air toxics emitted from multiple facilities. Criteria for selection are based on factors such as ecological relevance—for instance, a population of phytoplankton might serve as a foundational element in aquatic food webs; exposure potential, as exemplified by soil invertebrates with higher metabolic rates selected for their elevated ingestion rates relative to body weight; and sensitivity to these toxics, like certain benthic invertebrates that may be sensitive to heavy metals. These criteria serve to identify receptors that are most representative of the assessment endpoints for each community and class-specific guild. Consequently, the identified receptors serve as the key factors based upon which measures of effect assessments are conducted, utilizing toxicity benchmarks such as ambient water quality criteria and receptor-specific no-observed-adverse-effect levels.

For example, receptors for soil communities may include soil invertebrates and terrestrial plant communities. For surface water communities, receptors for surface water may extend to water invertebrates, phytoplankton, zooplankton, and fish communities.

### 2.7. Trophic-Level-Specific Exposure Assessment for Air Toxics

In trophic-level-specific exposure assessment for air toxics, the evaluation of exposure and toxicity varies depending on the ecological receptor’s position in the food chain. For lower-trophic-level organisms, exposure is assessed through direct uptake pathways from environmental media like soil, water, and sediment. In contrast, higher-trophic-level organisms are evaluated based on the ingestion of different organisms along with media contaminated with air toxics. These approaches, tailored for lower and higher trophic levels, respectively, make use of TRVs, which are either media-specific for lower trophic levels or dose-specific for higher trophic levels. The TRVs used in this assessment are derived from the peer-reviewed literature and validated databases, ensuring a rigorous and reliable basis for risk characterization.

This section describes the equations used for assessing exposure including:

#### 2.7.1. Daily Dose (DD) of Air Toxics Ingested by a Measurement Receptor

The calculation of this dose is key for assessing health risks across various trophic levels and can be determined using the following generic equation [2]:(1)$$DD=\sum IR_{F}\times C_{i}\times P_{i}\times F_{i}+\left(IR_{M}\times C_{M}\times P_{M}\right)$$
where *DD* represents the dose of air toxics ingested (mg air toxics/kg body weight-day), $IR_{F}$ is the measurement receptor animal or plant food item ingestion rate (kg/kg body weight-day), $C_{i}$ is the air toxics concentration in the *i*th animal or plant item (mg air toxics/kg), $P_{i}$ is the proportion of the *i*th food item that is contaminated (unitless), $F_{i}$ is the fraction of the diet consisting of animal or plant item *i* (unitless), $IR_{M}$ is the measurement receptor media ingestion rate (kg/kg body weight-day (bed sediment or soil) or L/kg body weight-day (water)), $C_{M}$ is the air toxics concentration in the medium (mg/kg (bed sediment or soil) or mg/L (water)), and $P_{M}$ is the proportion of ingested media that is contaminated (unitless). The calculation of this dose is adapted from methodologies developed by the US EPA, which are extended in this study to incorporate additional emission sources.

In calculating the daily dose within class-specific guilds, two distinct dietary scenarios are considered: Equal Diet and Exclusive Diet. In the Equal Diet, it is assumed that the measurement receptor consumes food items from different groups in equal proportions. The fraction of the diet consisting of each animal or plant item $F_{i}$ is thus equal to one divided by the total number of food item groups ingested. In contrast, the Exclusive Diet proposes that the measurement receptor’s diet is entirely composed of a single food item group at a time. Here, $F_{i}$ is set to one for the food item group, while $F_{i}$ values for all other food items are set to zero. This scenario is iteratively applied to each food item group, generating a range of potential exposure values based on exclusive diets.

#### 2.7.2. Bioconcentration Factor (BCF)

The concentrations of air toxics in phytoplankton, invertebrates, and rooted aquatic plants can be calculated from the steady-state bioconcentration factor (BCF), thereby enabling the evaluation of the dose ingested by the measurement receptor. The BCF is defined as the ratio of the concentration of air toxics in a biological food item to their concentration in the environmental media and can be calculated using Equation (2) [2]:(2)$$BCF=\frac{C_{i}}{C_{M}}$$
where *BCF* is the bioconcentration factor ((sediment, soil), or L/kg (water)) (unitless), $C_{i}$ is the air toxics concentration in ith animal or plant item (mg of air toxics/kg), and $C_{m}$ is the air toxics concentration in media (mg/kg (sediment, soil), or mg/L (water)).

#### 2.7.3. Food-Chain Multiplier (FCM)

A multiplier is applied to the bioconcentration factor (BCF) of air toxics to estimate the bioaccumulation factor (BAF), thereby accounting for the chemical’s accumulation up the food chain through predator–prey interactions. Air toxics with the greatest potential to biomagnify (defined by the authors as the increase in concentration of air toxics as they move upward through the different trophic levels) are highly lipophilic, exhibit low water solubility, and are resistant to metabolic breakdown [79]. The octanol–water partition coefficient serves as an important metric for understanding the bioaccumulative tendencies of air toxics, providing valuable insights into their transport, fate, and potential ecotoxicological impacts. The bioaccumulation factor can be calculated using Equations (3) and (4) [2]:(3)$$FCM=\frac{{BAF}_{l}}{K_{ow}}$$
(4)$$FCM=\frac{{BAF}_{l}}{{BCF}_{l}}$$
where *FCM* is the food-chain multiplier, ${BAF}_{l}$, lipid-normalized, represents the concentration of air toxics in aquatic organisms per unit volume of free chemical in the water, measured in units of L/kg, $K_{ow}$ is the octanol–water partition coefficient (L/kg), and ${BCF}_{l}$, lipid-normalized, quantifies the ratio of the air toxics concentration in aquatic organisms to their freely available concentration in the water, expressed in L/kg.

The FCM is used to estimate biomagnification and is equal to the ratio of the FCM of the measurement receptor divided by the predator’s FCM.

Table 4 provides a summary of key references where ecological risk assessors or researchers can find essential parameters for exposure assessment.

### 2.8. Toxicity Assessment for Air Toxics

The toxicity of air toxics is assessed through the identification of toxicity reference values (TRVs) that are specific both to the air toxics in question and to the measurement receptor under examination. The toxicity database used for these TRVs is the Ecotoxicology (ECOTOX) knowledgebase [83], which contains information on over 12,837 chemicals and 13,895 species. A TRV serves as a benchmark, specifically indicating a concentration or dose of air toxics that is not expected to cause observed adverse effects on ecologically relevant endpoints during chronic (long-term) exposure [2]. In ecological risk characterization, these values, tailored for lower-trophic-level communities, are specific to the media with which these organisms interact. For upper-trophic-level, class-specific guilds, the benchmarks are provided in terms of the dose ingested.

To illustrate the application of TRVs, consider the case of a mallard duck exposed to polycyclic aromatic hydrocarbons [84]. According to laboratory data, benchmarks indicating increased embryonic mortality were found at concentrations of 0.2 and 2.0 mg/kg egg. These concentrations serve as thresholds beyond which an increased risk of adverse effects on mallard duck embryos can be anticipated. This metric thus becomes crucial for evaluating the potential ecological risk posed by polycyclic aromatic hydrocarbons in avian communities.

TRVs serve as indispensable tools for systematically assessing and managing the potential risks that air toxics pose to both lower- and upper-trophic-level organisms, thereby contributing to the goal of ecological risk mitigation.

### 2.9. Ecological Risk Characterization

Ecological risk characterization encompasses both risk estimation and risk description. The quantification of risk estimation employs the quotient method, which yields an Ecological Screening Quotient (ESQ). This ESQ is calculated as a ratio wherein the estimated exposure level (EEL) of air toxics is divided by the TRV that is specific to both the air toxics and the measurement receptor. The formula is expressed as:(5)$$ESQ=\frac{EEL}{TRV}$$
where *ESQ* is the ecological screening quotient (unitless), EEL is the estimated exposure level to air toxics (mass air toxics/mass media (communities) or mass daily dose air toxics ingested/mass body weight-day (class-specific guilds)), and TRV is the air toxics’ toxicity reference value (mass air toxics/mass media (communities) or mass daily dose air toxics ingested/mass body weight-day (class-specific guilds)). This equation is adapted from methodologies developed by the US EPA [2].

In this study, the methodology is extended to incorporate additional emission sources such as glycol dehydration units and wastewater treatment plants. Furthermore, it is expanded to estimate the country-wide cumulative ecological risk from simultaneous exposure to air toxics, such as polycyclic aromatic hydrocarbons originating from multiple industrial sources.

To evaluate the potential cumulative risk posed by simultaneous exposure of a measurement receptor to multiple air toxics at a specific location, the air toxics-specific ESQ values should be aggregated to determine a cumulative ESQ, as presented in Equation (6):(6)$$Cumulative\;ESQ=\sum_{i=1}^{n}{ESQ}_{i}$$
where Cumulative ESQ is the total ecological screening quotient at a specific location (unitless), n is the number of air toxics under consideration, and ${ESQ}_{i}$ is the ecological screening quotient for the ith air toxic (unitless).

Following the estimation of risk, risk description articulates the scope and nature of potential ecological impacts, providing a critical interpretive context for ESQ values relative to regulatory benchmarks and ecological receptors. For example, the ESQ for fish continuously exposed to methylmercury in water bodies receiving industrial emissions is calculated at 3.5 × 10. Methylmercury is a bioaccumulative form of mercury that poses significant risks to aquatic life, particularly predatory fish, due to biomagnification through the food chain. The presence of methylmercury is attributed to the methylation of inorganic mercury emissions from industrial processes. Additionally, there is an inherent uncertainty in assuming that the fish diet is solely based on contaminated food sources; fish often have diverse diets that include a range of aquatic organisms and detritus. Nonetheless, due to the potential for methylmercury to bioaccumulate and the sensitivity of fish to its toxic effects, the exceedance of the target risk level warrants careful consideration and, if necessary, further investigation to confirm the levels of mercury in the aquatic environment and to assess the actual risk to the aquatic biota.

## 3. Analysis of Results

### 3.1. Case Study

Leveraging the methodology detailed above, this section presents the application of EHAM for a country-wide ecological health assessment in a case study of Kuwait, using data from 2017. Central to this assessment was the development of unique food web models, tailored to local ecosystems. These food webs, incorporating region-specific trophic interactions, enable a more accurate and localized risk assessment, important for understanding the ecological dynamics of Kuwait’s environment.

This study considers Kuwait’s distinct air quality zones, as shown in Figure 2, each with unique ecological characteristics and sensitivities. For example, coastal zones are integral to marine and shoreline ecosystems, while inland zones are crucial for desert flora and fauna. Further details pertaining to Kuwait’s air quality zones, the selection of air toxics emission sources for human health risk assessment, and data on modeled facilities and emission sources can be found in “Section 3.1. Case Study” of the authors’ companion paper on human health [1].

The four air toxics, benzene, formaldehyde, toluene, and benzo(a)pyrene, investigated in [1] are also the four air toxics investigated in this paper due to their common occurrence and well documented potency. A wide variety of emission sources, including combustion sources, glycol dehydrators, and wastewater treatment plant operations, were modeled. These sources were selected due to their notable contributions of air toxics, including the four air toxics assessed in this paper. For details on the entry mechanisms and affected trophic levels of these air toxics, readers are referred to Table 1. This approach supports an integrated perspective that facilitates a thorough assessment of ecological risks, thereby enhancing the understanding of air quality impacts in a wider ecological framework.

#### Food Web Models

For this study, unique food web models were developed, tailored to local ecosystems to achieve a country-wide ecological risk assessment for Kuwait. These models intricately mapped the interactions among species, organized by trophic levels and guilds to reflect the ecological structure of the region. Incorporating region-specific trophic interactions, they provided a more accurate and localized risk assessment. Drawing on the literature [85,86,87] to identify these biota, terrestrial species such as the red fox and the gerbil, and desert birds like the hoopoe, along with coastal species including the Indo-Pacific king mackerel, the greasy grouper, the crab plover, and the mudskipper, were grouped based on their feeding habits and roles within these ecosystems. This organization by trophic level—from primary producers like the Rhanterium epapposum plant to primary consumers and higher-level predators—enabled a detailed understanding of the ecological dynamics. These tailored models also employed food-chain multipliers, crucial for quantifying the biomagnification of air toxics such as benzo(a)pyrene across different trophic levels. By capturing both direct and indirect pathways of air toxics exposure, the models provided an in-depth understanding of the ecological risks associated with environmental contaminants in Kuwait.

### 3.2. Cumulative Risk Results

This section outlines the established target risk levels and presents findings related to the country-wide cumulative ecological risk, with a specific focus on community and guild-based measurement receptors within Kuwait’s three distinctive air quality zones: coastal, inland, and production. Within this assessment, an ESQ value of less than 1 is interpreted as indicating negligible ecological risk, suggesting that the level of air toxics exposure is below the threshold likely to cause adverse effects on local biota. Conversely, an ESQ of 1 or greater signals a potential risk, warranting further detailed investigation to ascertain the extent and implications of ecological impacts due to air toxics [2].

#### 3.2.1. Coastal Air Quality Zone

The cumulative ESQ analysis in the coastal zone reveals a range of ecological risks among different species, determined according to their specific diets and trophic interactions. Table 5 summarizes the cumulative ESQ values for seven receptor groups, providing insights into the variable ecological risks within this ecologically sensitive area.

In Table 5, when the diet type is listed as “Equal”, the ESQ is calculated based on an equal diet dose [2], meaning that the measurement receptor consumes food items from different groups in equal proportions. Conversely, when the diet type is anything other than “Equal”, it implies an exclusive diet, focusing on one particular food source. For instance, the ESQ value of 6.06 × 10^1^ for carnivorous birds under a diet of “Carnivorous Fish” indicates the risk assessment based on an exclusive diet of these fish.

Significantly, the data indicate a trend where measurement receptors at higher trophic levels tend to exhibit higher ESQ values compared to those at lower levels, likely due to the process of biomagnification. Carnivorous birds, for example, consistently show high ESQ values across different diets, suggesting considerable risk, particularly when their diet includes omnivorous birds and mammals. This pattern aligns with ecological principles where contaminants can accumulate in higher concentrations as one moves up the food chain, thus posing greater risks to predators at the top.

Carnivorous shorebirds also demonstrate elevated ESQ values, particularly when consuming benthic invertebrates and omnivorous birds, further indicating their vulnerability in the coastal ecosystem due to biomagnification. Conversely, carnivorous mammals exhibit a mixed risk profile, with varied ESQ values depending on their diet. Herbivorous birds and mammals, as well as omnivorous species, show variable ESQ values, reflecting a complex web of ecological interactions and risks associated with their specific feeding habits.

These findings highlight the need for policy measures aimed at reducing air toxics emissions, particularly those that can bioaccumulate. An example policy could involve stricter regulations on industrial emissions, coupled with enhanced monitoring and enforcement mechanisms. Such proactive measures would be important in safeguarding the coastal ecosystem and its diverse inhabitants from the escalating risks of biomagnification.

#### 3.2.2. Inland Air Quality Zone

This subsection examines the cumulative ecological risk in Kuwait’s inland air quality zone, with a specific focus on the assessment of ESQ values across various terrestrial species representative of this zone. Table 6 summarizes these findings, presenting the cumulative ESQ values for six receptor groups.

The analysis of Kuwait’s inland air quality zone presents a stark contrast to the coastal zone, with Table 6 showing all six receptor groups having cumulative ESQ values below 1.0. This uniformity indicates negligible ecological risk from air toxics across various species, including carnivorous, herbivorous, and omnivorous groups, irrespective of their trophic levels.

Notably, carnivorous birds and mammals exhibit extremely low ESQ values, suggesting minimal biomagnification impact. Similarly, herbivorous and omnivorous species display low ESQ values, highlighting the reduced influence of air toxics in this zone.

These findings suggest that current air quality standards in the inland zone are effectively protecting the terrestrial biota from air toxics exposure. The results emphasize the necessity of not only maintaining but also potentially strengthening these standards. Moreover, regular ecological risk assessments are recommended to ensure these standards continue to safeguard ecological health, adaptively managing and responding to evolving environmental and industrial challenges. This approach is important for the long-term protection and sustainability of these ecosystems.

#### 3.2.3. Production Air Quality Zone

This subsection analyzes the cumulative ecological risk within Kuwait’s production air quality zone. Table 7 presents these findings, summarizing the cumulative ESQ values for six receptor groups.

In Kuwait’s production air quality zone, Table 7 shows a consistent pattern of low ecological risk, with all six receptor groups exhibiting cumulative ESQ values well below 1.0. This uniformity across carnivorous, herbivorous, and omnivorous species indicates negligible risk from air toxics, a significant finding considering the industrial nature of this zone. Notably, both higher- and lower-trophic-level species show minimal biomagnification impact, suggesting effective air quality control in the production area. These encouraging results reinforce the effectiveness of current air quality regulations in this zone, emphasizing the importance of sustained and potentially enhanced environmental policies. Regular ecological risk assessments are vital to continually monitor and adapt these standards, ensuring that they respond effectively to the dynamic nature of industrial activities and environmental changes. This proactive approach to environmental management is crucial for maintaining ecological health and guiding future policy development in similar industrial contexts.

To provide a practical understanding of ecological risk from air toxics exposure, it is important to interpret the ESQ correctly. This quotient, derived as the ratio of EEL to TRV, as explained earlier in this paper, is not a probabilistic estimate. For example, an ESQ of 0.01 signifies that the exposure level is 100 times lower than the TRV, implying substantially reduced ecological risk, not a 1 in 100 chance of an adverse effect. An ESQ of 1.0 means that the exposure level is equal to the TRV, while an ESQ of 10 indicates an exposure level that is ten times higher than the TRV. For ESQ values greater than 1.0, a deeper examination into the specific toxicological profiles of the air toxics is necessary. This involves understanding the acute and chronic effects of these pollutants, their bioaccumulative potential across different species, and the extent of their impact on key ecological functions, such as nutrient cycling and predator–prey dynamics. An ESQ above 1.0 suggests potential ecological risk. However, higher ESQ values should not be interpreted linearly with risk magnitude—they indicate an increasing qualitative likelihood or severity of ecological risk from air toxics. Understanding these ESQ values is crucial for risk management, decision making, and the development of informed environmental policies. For instance, ESQs significantly greater than 1.0 could prompt further investigation, immediate mitigation efforts, or the adjustment of existing policies, whereas lower ESQs might indicate areas where current management and policy strategies are effective. Thus, these interpretations guide actions to proactively mitigate and manage ecological risks associated with air toxics, while also serving as a foundation for policy formulation and refinement.

### 3.3. Ecological Risk Driver Analysis

In the country-wide ecological risk assessment of Kuwait’s air quality zones, benzo(a)pyrene was identified as the risk driver in the coastal zone. This outcome aligns with findings from a companion human health risk study [1]. Benzo(a)pyrene’s notably higher bioconcentration factors, in comparison to other air toxics, signal its significant potential for bioaccumulation and subsequent ecological impact. This is particularly evident in the elevated cumulative ESQ values for carnivorous birds, calculated at 3.02 × 10^2^ for those with an equal diet and a cumulative ESQ reaching 1.01 × 10^3^ for those predominantly feeding on omnivorous birds. Both figures significantly exceed the established target risk levels. These values demonstrate benzo(a)pyrene’s capacity for bioaccumulation across different trophic levels within the food chain. This bioaccumulation highlights the substance’s ecological threat in the coastal zone, a habitat characterized by intricate interactions between marine and terrestrial species. Therefore, this analysis emphasizes the importance of assessing specific toxicological profiles in ecological risk assessments, especially for air toxics like benzo(a)pyrene that exhibit high bioconcentration factors.

### 3.4. Assumptions and Limitations of EHAM in the Kuwait Case Study

The country-wide ecological risk assessment of air toxics is based on several key assumptions and limitations, which are outlined as follows:(1)Bioavailability of Air Toxics: This study presumes that air toxics in food items and environmental media are fully bioavailable to ecological receptors. “Bioavailable” here means the fraction of a substance that is readily absorbed and can influence biological processes;(2)Presence of Sensitive Life Stages: This study assumes that the most sensitive life stages of measurement receptors are present within the assessment area, resulting in a more protective ecological health risk assessment;(3)Conservative Estimates: This study utilizes conservative estimates for body weights and food ingestion rates of measurement receptors;(4)Equal Exposure Across Species: This study operates under the assumption that each species within any given ecological guild or community is equally exposed to the air toxics;(5)Contaminated Food and Media: This study assumes that 100 percent of the food items and media ingested by receptors are contaminated, implying exclusive feeding within the assessment area;(6)Cross-Interaction of Air Toxics: The current assessment assumes independent effects of each air toxic, without considering potential cross-interactions, such as synergistic, antagonistic, or additive effects, that might occur when multiple pollutants coexist.

## 4. Conclusions

This paper introduces the Ecological Health Assessment Methodology (EHAM). To illustrate EHAM, a country-wide ecological risk assessment from air toxics is shown to be possible using Kuwait as the case study, revealing that it is not restricted to small regions. The key conclusions from this study are:(1)Substantial Ecological Risk in the Coastal Zone: This study demonstrates significant ecological risks in Kuwait’s coastal zone, particularly for carnivorous birds, with cumulative ESQ values reaching 3.12 × 10^3^. This highlights the substantial impact of air toxics in this ecologically sensitive area, driven largely by the process of biomagnification;(2)Negligible Risks in the Inland and Production Zones: In contrast, both the inland and production air quality zones exhibit negligible ecological risks, as indicated by the consistently low ESQ values across various species. This suggests the effective control of air toxics levels in these zones, protecting the ecological health of diverse species;(3)Benzo(a)pyrene as a Primary Ecological Risk Driver: Similar to findings in human health risk, benzo(a)pyrene is identified as a key risk driver in the coastal zone, due to its high bioconcentration factors and potential for bioaccumulation, underlining its significant ecological threat. This analysis highlights the importance of assessing specific toxicological profiles in ecological risk assessments, especially for air toxics like benzo(a)pyrene that exhibit high bioconcentration factors;(4)Development of Tailored Food Web Models for Local Ecosystems: This study’s unique models, customized for Kuwait’s ecosystems, offered a representative ecological risk assessment;(5)Importance of Diet Type and Trophic Level in Ecological Risk Assessments: This analysis highlights the importance of considering both the diet type and trophic level in assessing ecological risks due to air toxics exposure;(6)Need for Regular Ecological Risk Assessments: This study underscores the importance of conducting regular ecological risk assessments. These are important for adapting to environmental and industrial changes, ensuring that air quality standards continue to protect ecological health;(7)Informing Policy and Management Decisions: The findings and methodologies developed in this study offer valuable insights for environmental management strategies and policy decisions. These can guide actions to mitigate ecological risks, such as adjusting existing policies or implementing targeted mitigation efforts in high-risk areas.

## 5. Future Work

In future work, the authors plan to expand the EHAM analysis to include additional emission sources, such as fugitive and mobile sources, thereby broadening its ecological risk assessment scope. This expansion will be followed by a transition to a probabilistic approach, characterized by the development of a stochastic model incorporating distributions for variables like food ingestion rates of measurement receptors and fractions of diet. This evolution addresses the inherent variability and uncertainty in ecological risk assessments, enabling a more accurate representation of risk by capturing the range of possible outcomes in ecological systems and exposure scenarios. Continuing to explore the interactions and potential synergistic effects of different air toxics when multiple pollutants coexist, future studies will provide a more comprehensive understanding of their cumulative ecological impact. These advancements in EHAM will significantly contribute to more effective environmental management strategies and the ongoing development of ecological risk assessment methods.

## Figures and Tables

**Figure 1 toxics-12-00042-f001:**
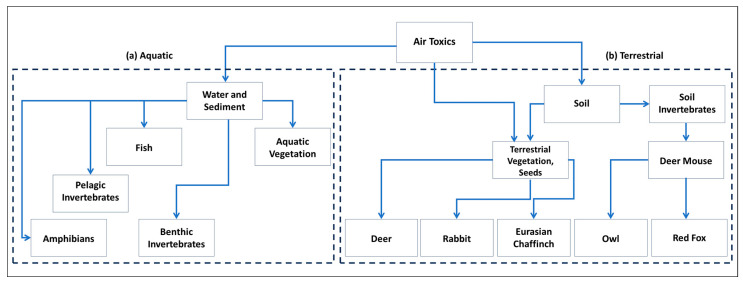
Example of transport pathways and fate of air toxics into example (**a**) aquatic and (**b**) terrestrial ecosystems.

**Figure 2 toxics-12-00042-f002:**
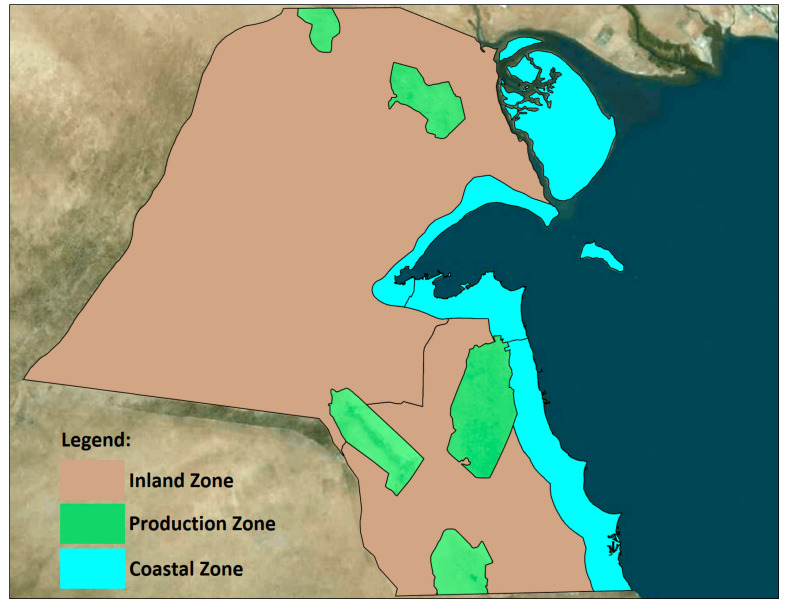
Kuwait air quality zones. Map data: Lakes Satellite.

**Table 1 toxics-12-00042-t001:** Comparison of entry mechanisms and affected trophic levels for air toxics in terrestrial and aquatic ecosystems [10,11,12,13,14,15,16,17,18,19,20,21,22,23,24,25,26,27,28,29,30,31,32,33,34,35,36,37,38].

	Terrestrial Ecosystems	Ref.	Aquatic Ecosystems	Ref.
Entry Mechanisms	Dry deposition	[10,11]	Rainfall	[21,22,23]
	Wet deposition	[10,11]	Tidal activities	[24,25,26]
	Direct inhalation	[12,13]	Direct uptake from water	[27,28,29]
	Plant–soil interactions	[14,15]	Watershed loading (surface runoff, point source discharge, atmospheric deposition, and riverine input)	[30,31,32]
Affected Trophic Levels	Herbivores	[16,17]	Benthic organisms	[33,34,35]
	Omnivores	[16,17]	Intermediate consumers	[36,37,38]
	Predators (via bioaccumulation)	[16,17]	Larger aquatic predators (via bioaccumulation)	[16,17,24]
Inter-ecosystem Interactions	Watershed loading affects adjacent aquatic ecosystems	[18,19,20]	Watershed loading affects adjacent terrestrial ecosystems	[18,19,20]

**Table 2 toxics-12-00042-t002:** Examples of significant health impacts of air toxics on select organisms and their documented responses.

Ecological Health Impact	Ref.
Amphibian (Common Frog): Increased embryonic mortality rate	[39]
Bird (Pied Flycatchers): Habitat degradation; impaired reproductive success	[40,41]
Fish (African Catfish): Decreased sperm motility	[42]
Fish (Common Carp): Increased lipid peroxidation; DNA damage; elevated micronuclei frequency in blood, gill, and liver	[43]
Fish (European Perch): Decline in sperm performance; sperm structural damage affecting flagella, plasma membrane, and axoneme; decreased sperm ATP content leading to inhibited motility; damage to sperm’s midpiece and mitochondria during prolonged exposure; disruption in sperm activation due to plasma membrane damage	[44]
Fish (Lake Trout, Largemouth Bass, Northern Pike, Walleye): Behavioral alterations; changes in gene expression; impacts on growth	[45]
Fish (Norwegian Waters): Induction of phase-I enzymes; development of DNA adducts; formation of neoplastic lesions	[46]
Fish (Peacock Blenny): Circulatory issues in gills; liver damage comprising reduction in cell size, loss of cellular integrity, and architectural disruptions	[47]
Fish (Trahira): Electrophysiological changes in retinal horizontal cells	[48]
Fish (Zebrafish): Changes in lipid metabolism and cellular transport; gonadal damage; oxidative stress in the testis; altered sex hormone levels impairing reproduction	[49,50]
Mammal (Mink): Reproductive failure	[51]
Mesozooplankton: Accumulation of air toxics in tissues, fecal pellets, and eggs; potential transfer to higher trophic levels affecting pelagic and benthic communities	[52]
Phytoplankton: Disruption to growth and photosynthesis; induction of oxidative stress	[53,54]

**Table 3 toxics-12-00042-t003:** Examples of past ecological risk assessments, along with this present work, with source, pathway, and receiver characterization.

#	Literature (Year)	Country-Wide	Source	Pathway	Receiver	Ref.
1	A Procedure for Performing Population-Level Ecological Risk Assessments (2000)	No	Land/water	Land to local ecosystems	Quail, shrew, fish	[57]
2	Potential Long-Term Ecological Impacts Caused by Disturbance of Contaminated Sediments: A Case Study (2002)	No	Water	Release from disturbed sediments to water column	Mummichog, blue crab, polychaete, striped bass	[58]
3	An Ecological Risk Assessment for Spinosad Use on Cotton (2002)	No	Land	Terrestrial insecticide application to cotton crops	Groundwater, mourning dove, field sparrow, blue tit, bees, bluegill, rainbow trout, carp, grass shrimp, eastern oyster, freshwater diatom, Sharkey soil	[59]
4	Ecological Risk Assessment of Contaminated Soils Through Direct Toxicity Assessment (2005)	No	Land	Leaching of contaminants from soil	Plants, earthworm, Daphnia, algae, fish	[60]
5	Probabilistic Ecological Risk Assessment of 1,2,4-Trichlorobenzene at a Former Industrial Contaminated Site (2005)	No	Land	Leaching from soil to groundwater	Aquatic invertebrates, microbial soil population, plants	[61]
6	A Screening-Level Assessment of Lead, Cadmium, and Zinc in Fish and Crayfish from Northeastern Oklahoma, USA (2006)	No	Water	Runoff from land to water bodies	Humans, carp, catfish, bass, crappie, carnivorous wildlife, crayfish	[62]
7	Anthropogenic Input of Selected Heavy Metals in the Aquatic Sediments of Hochiminh City, Vietnam (2006)	No	Water	Industry and agricultural runoff	Aquatic sediments in rivers and canals	[63]
8	Potential Importance of Inhalation Exposures for Wildlife Using Screening-Level Ecological Risk Assessment (2006)	No	Air	Airborne contaminants inhaled by wildlife	Small mammals	[64]
9	Environmental Risk Assessment of Pharmaceutical Residues in Wastewater Effluents, Surface Waters, and Sediments (2006)	No	Water	Wastewater and surface water contamination	Aquatic organisms	[65]
10	Bioconcentration of Dioxins and Furans in Vegetation (2007)	No	Land	Soil-to-vegetation contamination	Vegetation including grass and clover	[66]
11	Predicting Pesticide Environmental Risk in Intensive Agricultural Areas. II: Screening Level Risk Assessment of Complex Mixtures in Surface Waters (2009)	No	Land	Pesticide drift and runoff	Algae, Daphnia, fish	[67]
12	Evidence of impacts from DDT in pelican, cormorant, stork, and egret eggs from KwaZulu-Natal, South Africa (2019)	No	Land	Aerial transport and aquatic ecosystems	African openbill, pelican, molluscan, piscivorous birds, pink-backed pelican, egrets, white-breasted cormorant	[68]
13	DSS-OSM: An Integrated Decision Support System for Offshore Oil Spill Management (2021)	No	Water	Spread of oil in marine environments	Marine organisms	[69]
14	This Work (2023)	Yes	Air	Atmospheric dispersion and deposition of air toxics	Terrestrial and aquatic species across various trophic levels	

**Table 4 toxics-12-00042-t004:** Summary of key references for parameters used in exposure assessment.

Parameters	Sources	Remarks
Ingestion Rates for Measurement Receptors	US EPA Wildlife Exposure Factors Handbook [80]	Data on ingestion rates across species.
Food-chain Multiplier	Great Lakes Water Quality Initiative Technical Support Document for the Procedure to Determine Bioaccumulation Factors [81]	Provides standardized FCM values.
Bioaccumulation Factor Model	Gobas [82]	Models for calculating BAF.

**Table 5 toxics-12-00042-t005:** Summary of cumulative ESQ values for measurement receptors in the coastal zone.

#	Receptor Group	Diet	Cumulative ESQ (Unitless)	ESQ Above 1.0 (Y = Yes/N = No)
1	Carnivorous Bird	Equal	3.02 × 10^2^	Y
		Exclusive (Detailed Breakdown Below)	-	-
		Carnivorous Fish	6.06 × 10^1^	Y
		Herbivorous Bird	1.15 × 10^1^	Y
		Herbivorous Mammal	1.18 × 10^1^	Y
		Omnivorous Bird	1.01 × 10^3^	Y
		Omnivorous Fish	4.15 × 10^1^	Y
		Omnivorous Mammal	6.77 × 10^2^	Y
2	Carnivorous Mammal	Equal	2.66 × 10	Y
		Exclusive (Detailed Breakdown Below)	-	-
		Carnivorous Fish	4.65 × 10^−1^	N
		Herbivorous Bird	1.84 × 10^−2^	N
		Herbivorous Mammal	2.11 × 10^−2^	N
		Omnivorous Bird	9.07 × 10	Y
		Omnivorous Fish	2.91 × 10^−1^	N
		Omnivorous Mammal	6.06 × 10	Y
3	Carnivorous Shorebird	Equal	9.04 × 10^2^	Y
		Exclusive (Detailed Breakdown Below)	-	-
		Benthic Invertebrates	1.08 × 10^3^	Y
		Omnivorous Bird	3.12 × 10^3^	Y
		Omnivorous Fish	1.40 × 10^2^	Y
		Planktivore Fish	5.59 × 10^1^	Y
		Water Invertebrates	1.27 × 10^2^	Y
		Benthic Invertebrates	1.08 × 10^3^	Y
4	Herbivorous Bird	Equal	1.87 × 10^1^	Y
		Exclusive (Detailed Breakdown Below)	-	-
		Algae	3.30 × 10^1^	Y
		Aquatic Vegetation	4.39 × 10	Y
5	Herbivorous Mammal	Equal	6.42 × 10^−1^	N
		Exclusive (Detailed Breakdown Below)	-	-
		Algae	1.17 × 10	Y
		Aquatic Vegetation	1.11 × 10^−1^	N
6	Omnivorous Bird	Equal	5.15 × 10^2^	Y
		Exclusive (Detailed Breakdown Below)	-	-
		Algae	1.66 × 10^2^	Y
		Aquatic Vegetation	3.32 × 10^1^	Y
		Benthic Invertebrates	1.71 × 10^3^	Y
		Water Invertebrates	1.51 × 10^2^	Y
7	Omnivorous Mammal	Equal	1.59 × 10	Y
		Exclusive (Detailed Breakdown Below)	-	-
		Algae	7.11 × 10^−1^	N
		Aquatic Vegetation	8.03 × 10^−2^	N
		Benthic Invertebrates	8.04 × 10	Y
		Herbivorous Bird	2.93 × 10^−2^	N
		Herbivorous Mammal	3.66 × 10^−2^	N
		Water Invertebrates	6.38 × 10^−1^	N

**Table 6 toxics-12-00042-t006:** Summary of cumulative ESQ values for measurement receptors in the inland zone.

#	Receptor Group	Diet	Cumulative ESQ (Unitless)	ESQ Above 1.0 (Y = Yes/N = No)
1	Carnivorous Bird	Equal	4.55 × 10^−6^	N
		Exclusive (Detailed Breakdown Below)	-	-
		Herbivorous Bird	1.49 × 10^−7^	N
		Herbivorous Mammal	1.92 × 10^−7^	N
		Omnivorous Bird	1.16 × 10^−5^	N
		Omnivorous Mammal	6.25 × 10^−6^	N
2	Carnivorous Mammal	Equal	3.91 × 10^−8^	N
		Exclusive (Detailed Breakdown Below)	-	-
		Herbivorous Bird	2.60 × 10^−9^	N
		Herbivorous Mammal	2.96 × 10^−9^	N
		Omnivorous Bird	9.77 × 10^−8^	N
		Omnivorous Mammal	5.31 × 10^−8^	N
3	Herbivorous Bird	Equal	4.28 × 10^−6^	N
		Exclusive (Detailed Breakdown Below)	-	-
		Terrestrial Plant	4.28 × 10^−6^	N
4	Herbivorous Mammal	Equal	7.00 × 10^−8^	N
		Exclusive (Detailed Breakdown Below)	-	-
		Terrestrial Plant	7.00 × 10^−8^	N
5	Omnivorous Bird	Equal	3.68 × 10^−6^	N
		Exclusive (Detailed Breakdown Below)	-	-
		Terrestrial Invertebrates	2.71 × 10^−6^	N
		Terrestrial Plant	4.65 × 10^−6^	N
6	Omnivorous Mammal	Equal	2.82 × 10^−8^	N
		Exclusive (Detailed Breakdown Below)	-	-
		Herbivorous Bird	2.71 × 10^−9^	N
		Herbivorous Mammal	3.37 × 10^−9^	N
		Terrestrial Invertebrates	3.41 × 10^−8^	N
		Terrestrial Plant	7.26 × 10^−8^	N

**Table 7 toxics-12-00042-t007:** Summary of cumulative ESQ values for measurement receptors in the production zone.

#	Receptor Group	Diet	Cumulative ESQ (Unitless)	ESQ Above 1.0 (Y = Yes/N = No)
1	Carnivorous Bird	Equal	7.09 × 10^−7^	N
		Exclusive (Detailed Breakdown Below)	-	-
		Herbivorous Bird	2.37 × 10^−8^	N
		Herbivorous Mammal	3.11 × 10^−8^	N
		Omnivorous Bird	1.81 × 10^−6^	N
		Omnivorous Mammal	9.76 × 10^−7^	N
2	Carnivorous Mammal	Equal	6.09 × 10^−9^	N
		Exclusive (Detailed Breakdown Below)	-	-
		Herbivorous Bird	4.09 × 10^−10^	N
		Herbivorous Mammal	4.70 × 10^−10^	N
		Omnivorous Bird	1.52 × 10^−8^	N
		Omnivorous Mammal	8.30 × 10^−9^	N
3	Herbivorous Bird	Equal	7.17 × 10^−7^	N
		Exclusive (Detailed Breakdown Below)	-	-
		Terrestrial Plant	7.17 × 10^−7^	N
4	Herbivorous Mammal	Equal	1.18 × 10^−8^	N
		Exclusive (Detailed Breakdown Below)	-	-
		Terrestrial Plant	1.18 × 10^−8^	N
5	Omnivorous Bird	Equal	5.97 × 10^−7^	N
		Exclusive (Detailed Breakdown Below)	-	-
		Terrestrial Invertebrates	4.20 × 10^−7^	N
		Terrestrial Plant	7.75 × 10^−7^	N
6	Omnivorous Mammal	Equal	2.82 × 10^−8^	N
		Exclusive (Detailed Breakdown Below)	-	-
		Herbivorous Bird	4.64 × 10^−9^	N
		Herbivorous Mammal	4.30 × 10^−10^	N
		Terrestrial Invertebrates	5.43 × 10^−10^	N
		Terrestrial Plant	5.31 × 10^−9^	N

## Data Availability

A subset of the data is available upon request from the corresponding author, due to confidentiality constraints on certain portions of the dataset.

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
