# Peer review of "Country-Wide Ecological Health Assessment Methodology for Air Toxics: Bridging Gaps in Ecosystem Impact Understanding and Policy Foundations"

_toxics, 2024, doi:10.3390/toxics12010042_

Round 1
Reviewer 1 Report
Comments and Suggestions for Authors
The document titled "Country-Wide Ecological Health Assessment Methodology for Air Toxics: Bridging Gaps in Ecosystem Impact Understanding and Policy Foundations" introduces a novel methodology, named Ecological Health Assessment Methodology (EHAM), for assessing national-level ecological risks posed by air toxics. EHAM stands out for its capability to conduct both comprehensive country-wide and detailed regional ecological risk assessments, covering both terrestrial and aquatic ecosystems. This approach surpasses traditional methods used, for example, by the United States Environmental Protection Agency (US EPA), by providing a more versatile assessment of different emission sources and their ecological repercussions.
The paper primarily focuses on industrial sources of pollution, while acknowledging that there are other significant emission sources, like fugitive emissions and mobile sources, that are not included in EHAM. However, the framework of EHAM is flexible enough to be adapted to incorporate these additional sources. Using Kuwait as a case study, the authors adapted air dispersion models and deposition simulations originally conducted for human health risk assessments to estimate ecological risks associated with air toxics.
EHAM employs accepted methods for emissions estimation and air dispersion modeling, like the AERMOD model, and focuses on steps such as developing a national emissions inventory, modeling air dispersion and deposition, characterizing exposure settings, developing habitat-specific food webs, selecting assessment endpoints within each trophic level of the food web, identifying measurement receptors, assessing exposure via direct uptake and ingestion, and evaluating the toxicity of air toxics.
- Incorporate Additional Emission Sources: Consider integrating fugitive emissions and mobile sources into the EHAM model to provide a more comprehensive assessment of pollution sources.
- Further Explore Complex Ecosystem Interactions: Delve deeper into the complex interactions between various terrestrial and aquatic ecosystems and how air toxics affect these dynamics.
- Enrich the introductory part by talking about other methodologies used to evaluate the impact of air pollutants on human health, inserting studies such as: Comparison Process of Blood Heavy Metals Absorption Linked to Measured Air Quality Data in Areas with High and Low Environmental Impact in the analyses. 10.3390/pr10071409
- Geographic Expansion of Case Study: Use additional case studies beyond Kuwait to examine the applicability and effectiveness of EHAM in different geographic and environmental contexts.
- Deepen Analysis of Results: Provide a more detailed analysis of findings, particularly in terms of how specific discoveries might influence environmental policies and practices.
- Assess Long-Term Impact: Conduct a study on the long-term impact of air toxics on ecosystems to better understand the long-term consequences and guide future planning.
Reviewer 2 Report
Comments and Suggestions for Authors
This paper is very interesting and merits full consideration by the journal. This is especially true for the possible implications of politics targeting emission control in the area under study. In my opinion, most of this paper is related to heavy metals. Although this is an important issue, other pollutants merit citations. For instance, persistent organics (POP) are scarcely cited.
Another point to consider concerns the values of the cumulative ESQ. Is this affected by a cross-interaction between the many pollutants found in the environment?
Because an air pollution model was run in the area, it would be interesting to report detailed information about the pollution distribution in the study area.
In line 112, the statement should be linked to the existing air quality standard for toxic emissions. The intake of food increases the risk of exposure. Additional comments on these issues are recommended.
Lines 188–189 provide information regarding the uncertainty of the emission estimates.
In line 261, TL3 and Tl4 are explained: move this definition to paragraph starting at line 246, where TL1 and TL2 are also given.
Finally, increase the font size in Fig. 1
Reviewer 3 Report
Comments and Suggestions for Authors
This is an interesting paper on an important matter.
Some English language corrections are necessary eg
you state . A previous study [1] recommended a country-wide
what country? why should it be country wide?
replace
Although this paper 46 focuses on estimating ecological risks primarily from industrial sources, it should be noted 47 that EHAM does not include other significant emission sources, such as fugitive emissions 48 or mobile sources
with
This paper 46 focuses on estimating ecological risks primarily from industrial sources, and it should be noted 47 that at this point EHAM does not include other significant emission sources, such as fugitive emissions 48 or mobile sources
replace
In summary, the introduction of EHAM provides a structured, yet adaptable, frame- 86 work for assessing the country-wide ecological risks posed by air toxics
with
In summary, EHAM as described here may provide a structured and adaptable, frame- 86 work for assessing the ecological risks posed by air toxics at country-wide level
and so on...
2) In lines 59-68 you talk aboyt ecosystems impotance for ecosystems such as wetlands and forests however your case study is a desert and a coastal ecosystem (?). It is more appropriate to focus on these ecosystems then
3) I am not convinced of the usefulness of Fig 1. You showed the mechanisms of path entries in Table 1 why you also need Fig 1. It offers nothing more except it has some examples of biota. Please elaborate on its use
4) the same about Table 2. I have not understood are you going to use these endpoints in your model also? or are they just for literature reasons? what is the validity of these studies? are they just so few? what were the keyworrds etc that you used to find them? at this point the selection does not seem scientific enough
5) the same about Table 3 what is the use of this table since it clearly does not focus on air pollution but on all kinds of sources. There is a robust risk assessment scheme from EFSA especially for plant protection products that is evolving and it deals with all compartments and all biota (terrestrial aquatic, etc) which is widely acknowledged. Maybe it is not so specific for air pollution so there is a gap but it makes no sense to me to show some research on ecological risk assessment while there are clear guidelines at least in EU
6) As I understand there are two main parts of the methods: one that you decsribe the method and its inputs and another one where you give your case study. In the first part lines 175-455 please decribe in a more consice way your inputs and reduce the text found here. I cannot understand at all the use of table 4 and 5-are they used in the model at all?
7) I am not sure how Daily Dose (DD) of Air Toxics Ingested by a Measurement Receptor is used-it is used in relation to what toxicity database? I am not sure how the the threshold reference value (TRV) was assessed? is it based on inhalation or on ingestion? if the TRV is already designed by USEPA what is the additional novelty made by your model? all these should be clearly explained
8) the case study you propose is really interesting but I think it gives minimum data on your inputs. what exactly is the geographic location of the case study? all the country including cities? the air pollution inventory from when until when? what pollutants it contains? the biota examples used in what ecosystems they belong etc. at this point you give mainly results and not the input data you used. at this point there are too few data given to evaluate the validity of the model
Comments on the Quality of English Languagemoderate corrections
Round 2
Reviewer 3 Report
Comments and Suggestions for Authors
ok
Comments on the Quality of English Languageneeds editing